# Analysis of *BRCA1* and *RAD51C* Promoter Methylation in Italian Families at High-Risk of Breast and Ovarian Cancer

**DOI:** 10.3390/cancers12040910

**Published:** 2020-04-08

**Authors:** Silvia Tabano, Jacopo Azzollini, Chiara Pesenti, Sara Lovati, Jole Costanza, Laura Fontana, Bernard Peissel, Monica Miozzo, Siranoush Manoukian

**Affiliations:** 1Medical Genetics, Department of Pathophysiology and Transplantation, Università degli Studi di Milano, 20122 Milano, Italy; silvia.tabano@unimi.it (S.T.); monica.miozzo@unimi.it (M.M.); 2Laboratory of Medical Genetics, IRCCS Ca’ Granda, Ospedale Maggiore Policlinico Milano, 20122 Milano, Italy; sara.lovati@studenti.unimi.it; 3Unit of Medical Genetics, Department of Medical Oncology and Hematology, Fondazione IRCCS Istituto Nazionale dei Tumori, 20133 Milano, Italy; Jacopo.Azzollini@istitutotumori.mi.it (J.A.); bernard.peissel@istitutotumori.mi.it (B.P.); 4Oncology Department, Istituto di Ricerche Farmacologiche Mario Negri, 20156 Milano, Italy; chiara.pesenti@marionegri.it; 5Unit of Research Laboratories coordination, IRCCS Ca’ Granda, Ospedale Maggiore Policlinico Milano, 20122 Milano, Italy; jole.costanza@policlinico.mi.it (J.C.); laura.fontana@unimi.it (L.F.)

**Keywords:** germline epigenetic defects, *BRCA1*, *RAD51C*, promoter methylation, breast carcinoma, ovarian carcinoma

## Abstract

Previous studies on breast and ovarian carcinoma (BC and OC) revealed constitutional *BRCA1* and *RAD51C* promoter hypermethylation as epigenetic alterations leading to tumor predisposition. Nevertheless, the impact of epimutations at these genes is still debated. One hundred and eight women affected by BC, OC, or both and considered at very high risk of carrying *BRCA1* germline mutations were studied. All samples were negative for pathogenic variants or variants of uncertain significance at *BRCA* testing. Quantitative *BRCA1* and *RAD51C* promoter methylation analyses were performed by Epityper mass spectrometry on peripheral blood samples and results were compared with those in controls. All the 108 analyzed cases showed methylation levels at the *BRCA1/RAD51C* promoter comparable with controls. Mean methylation levels (± stdev) at the *BRCA1* promoter were 4.3% (± 1.4%) and 4.4% (± 1.4%) in controls and patients, respectively (*p* > 0.05; *t*-test); mean methylation levels (± stdev) at the *RAD51C* promoter were 4.3% (± 0.9%) and 3.7% (± 0.9%) in controls and patients, respectively (*p* > 0.05; *t*-test). Based on these observations; the analysis of constitutional methylation at promoters of these genes does not seem to substantially improve the definition of cancer risks in patients. These data support the idea that epimutations represent a very rare event in high-risk BC/OC populations.

## 1. Introduction

Epigenetic alterations, i.e., epimutations, are an emerging mechanism that plays a pivotal role in carcinogenesis. Several studies demonstrated that defects in epigenetic markers, such as modifications of CpG methylation at gene promoters, result in transcriptional silencing of active genes or activation of silent genes, causing effects similar to those of gene mutations. Particularly, DNA methylation at CpG sites located in promoter regions of tumor-suppressor genes is a well-described event that could be present: (i) only in tumor cells, i.e., somatic epimutations, (ii) at the germline level, with evidence of inheritance, i.e., linked to *cis*-acting mutations [1,2] or, finally, (iii) in multiple tissues of different embryonic origin with no evidence of inheritance and often in a mosaic state, i.e., constitutional epimutations.

Regarding breast and ovarian cancer (BC and OC), mutations in *BRCA1*, along with *BRCA2* and few other genes, are responsible for only a fraction of familial cases. Moreover, somatic epimutations in these genes are nowadays largely studied to understand not only their involvement in tumorigenesis but also their effects on therapy response, such as at cisplatin and PARPi [3,4]. The study of germline and constitutional epimutations in BC and OC related genes is a relatively novel research field. Recently, Evans and colleagues reported the first case of germline *BRCA1* methylation in two families affected by Hereditary Breast and Ovarian Cancer Syndrome. The *BRCA1* promoter resulted hemi-methylation in all cells and it was linked to a *cis*-acting 5’UTR promoter variant [5]. Constitutional *BRCA1* methylation was analyzed by different groups with multiple technical approaches and it has been demonstrated to be associated with an increased risk of developing BC and/or OC [6,7,8,9,10,11]. Indeed, constitutional *BRCA1* methylation has been related to early onset of the disease and to basal-like phenotypes in case of BC, especially the triple negative (TN) subtype, and high-grade serous histotypes in the case of OC. Along with *BRCA1*, other BC/OC-related genes have been investigated. Among these, *RAD51C* was found constitutively hypermethylated in sporadic BC, as showed by Hansmann and colleagues [10], who identified hypermethylation at both *BRCA1* and *RAD51C* promoters in 1.4% and 0.5% of high-risk cancer patients, respectively. The authors noted that epimutations were present in 5.5%, (2/37) sporadic early-onset BC patients, 0.9% (4/460) familial BC patients, and in 10% (4/39) of OC patients. Moreover, epimutations were confined to one of the two parental alleles always at the mosaic level in tissues of different embryonic origins. This indicated that they occurred in a single cell relatively early during the embryonic development. This model was further supported by recent studies demonstrating that the majority of constitutional *BRCA1* methylation events arose early during development and seemed to remain quite stable during life and potentially were inherited from mother to daughter [12,13]. This novel hypothesis of inheritance is controversial and needs further studies to be completely elucidated [13].

Apart from the etiopathogenesis of this phenomenon, its prevalence is still debated. Depending on the studies, it ranges from less than 1% to 15%. This lack of homogeneity might be due to different composition of the patients’ cohorts as well as to the different nature of the technical approaches applied to profile methylation, which were not always quantitative. Defining the prevalence of constitutional epimutations in BC and OC might improve screening strategies for identifying women at increased risk who may benefit from tailored preventive options.

Our group recently reported constitutional *BRCA1* methylation in one out of 154 selected isolated early-onset BC patients, exploiting an accurate quantitative approach and establishing strict analysis parameters [11].

Based on the previous findings, we tried to clarify the prevalence of constitutional epimutations in BC and OC predisposition by analyzing the methylation status of *BRCA1* and *RAD51C* promoters in 108 women affected by BC and/or OC, with high a priori risk of harboring pathogenetic variants at *BRCA1* and who tested negative for *BRCA* sequence variants.

## 2. Results

### BRCA1 and RAD51 Methylation Levels

As shown in Figure 1a, none of the 108 analyzed cases exceeded the fixed minimum hypermethylation threshold defined in controls (threshold values were 13.6% and 12.1% for *BRCA1* and *RAD51C*, respectively). In detail, mean methylation levels (± stdev) of the *BRCA1* promoter region were 4.3% (± 1.4%) and 4.4% (± 1.4%) in controls and patients, respectively (*p* > 0.05, *t*-test); mean methylation levels (± stdev) of the *RAD51C* promoter region were 4.3% (± 0.9%) and 3.7% (± 0.9%) in controls and patients, respectively (*p* > 0.05, *t*-test). These overall data are consistent with the absence, in the investigated population, of germline or constitutive mosaic epigenetic alterations, resulting in *BRCA1* or *RAD51C* promoter hypermethylation.

We also investigated possible differences in methylation levels among clinical subgroups, without finding significant differences. In particular, patients affected with BC-only showed methylation levels at *BRCA1* and *RAD51C* comparable with those affected with OC-only (*p* > 0.05, *t*-test). In detail, *BRCA1* mean methylation was 4.3% (± 0.93%) and 3.9% (± 0.97%) in patients with BC- or OC-only, respectively. *RAD51C* mean methylation was 3.7% (± 0.91%) and 3.5% (± 0.56%) in patients with BC- or OC-only, respectively (Figure 1b). No differences were noted among different histological subtypes of both BC and OC, albeit the number of cases in each subgroup might be too small to highlight slight discrepancies.

## 3. Discussion

This study evaluated the presence of DNA hypermethylation at promoter regions of *BRCA1* and *RAD51C* genes in peripheral blood lymphocytes from women affected by BC and/or OC who tested negative at the *BRCA* analysis, although they were at high risk to harbor pathogenic variants. The rationale of the study was based on previous findings on the possible influence of germline hypermethylation of cancer predisposing gene promoters [1,2] in a few conditions including breast cancer [14].

The *BRCA1* and *RAD51C* genes were selected as potential candidate genes based on previous studies and on the phenotype of patients included in the study. While *BRCA1* is the most recognized predisposing gene for BC and OC, *RAD51C* was first identified as a putative cancer-predisposing gene in BC/OC families in 2010 [15]. Subsequently, it was confirmed to be associated with an increased risk of OC and, only recently, of BC, in particular of the TN type [16,17]. Moreover, to date somatic epimutations in both BC [5,10,18,19] and OC [10,19,20] have been identified only at these two genes.

In our study, none of the patients exhibited increased methylation levels at *BRCA1* and *RAD51C* promoters, with respect to the control population. No statistical differences were detected among specific subgroups, in particular patients affected with BC only and those affected with OC only.

Although this result might be burdened by the limited size of our cohort, our approach, based on highly selective inclusion criteria, should have maximized the chances of identifying potentially causative epimutations. Therefore, we hypothesize that epimutations, either germline or constitutional, represent a very rare event in women with BC and/or OC who tested negative at the *BRCA* analysis, although they were at high risk to harbor pathogenic variants. This finding also supports previous reports [21,22], highlighting the absence of germline epimutations at *BRCA1* or other BC/OC-predisposing genes in similar cohorts. Moreover, this result is in line with our previous study on isolated early-onset BC patients, which demonstrated the extremely low frequency of constitutional epiphenomena in a different subgroup of patients considered to be at increased risk to carry cancer predisposing alterations [11].

In the attempt to clarify the importance of epimutations, instead of genetic variants, in cancer predisposition genes, a number of technical limitations should be taken into account: (i) blood might not be the ideal tissue, since the hypermethylation could be present in a mosaic condition and blood may be not involved in the alteration; (ii) the sensitivity of the methylation detection may not completely exclude the presence of constitutional mosaic epimutations. In our analyses the lower threshold of hypermethylation was relatively high (i.e., +0.5 above the one-sided 95% bootstrap confidence interval) and archival tumor samples were not available for this study, therefore we were able to investigate germline epimutations though we could not completely rule out the presence of mosaic epimutations. Nevertheless, a mosaic hypermethylation would not fit in with the family history of most of our patients, who reported multiple BC/OC affected family members.

Additional factors, not assessed in the present study, including age or chemotherapy treatment, might affect methylation levels in blood. Notwithstanding, we previously investigated some of these factors on a cohort of isolated early onset BC patients and found that blood methylation at these loci was stable regardless of age at blood withdrawal or previous chemotherapy [11].

## 4. Materials and Methods

### 4.1. Study Subjects

A total of 108 patients were selected among individuals who underwent genetic counseling and testing at Fondazione IRCCS Istituto Nazionale dei Tumori di Milano (INT). All patients enrolled in the study were affected by either BC, OC, or both and were considered at very high risk of carrying a *BRCA1* pathogenic variants based on personal and/or family history. Patients who fulfilled one of the four criteria reported in Table 1 were included in the study.

All the BC cases were either invasive or ductal carcinomas in situ (DCIS). All the OC included in the analysis were not of the mucinous type or low grade. A “high mutation probability” was defined as ≥30% according to the BRCAPRO model (CaGene software version 5.1; PMID 9443863, http://www4.utsouthwestern.edu/breasthealth/cagene/) or either ≥30% for *BRCA1/BRCA2* or ≥60% for *BRCA1/BRCA2* plus a hypothetical third gene, which would account for all the other BC-associated genes in *BRCA*-negative families, according to the case-only-study, COS software [23]. All the selected patients were negative for pathogenic variants (including large deletions/duplications) or variants of uncertain significance (VUS) in *BRCA1* and *BRCA2*.

All patients underwent the methylation analysis on peripheral blood leukocytes (PBLs). Eighty-nine patients developed BC, with a median age at onset of 34 years (range 24–71 years). Among these women, 11 developed ipsilateral BC (median age 40, range 36–61), 23 contralateral BC (median age 45, range 33–65), and 6 OC (median age 54.5, range 40–78). Among the 19 patients who developed OC, but not BC, the median age at onset was 61 (range 32–76). The age at blood withdrawal ranged from 27 to 78 years (median 46). The main histopathological features of BC and OC cases included in the study are reported in Table 2 and Table 3, respectively.

The analysis was also carried out in a control population of 60 women, selected among healthy blood donors at the Immunohematology and Transfusion Medicine Service of our Institution. The age at blood withdrawal in controls ranged from 21 to 59 years (median 41).

All participants provided a signed informed consent for the use of their biological samples and data for research purposes. The investigations were conducted in accordance with the Declaration of Helsinki and the study was approved by the Ethics Committee of INT (approval code INT 171/15).

### 4.2. DNA Extraction and Bisulfite Conversion

Genomic DNA was purified from PBLs using the MagCore Super instrument and the MagCore Genomic DNA Whole Blood Kit (Diatech Lab Line, JEsi, Italy, CE-IVD). Bisulfite conversion was conducted on 500 ng of DNA using the EZ Direct DNA Methylation Kit (Zymo Research, Irvine, CA, USA), following the manufacturer’s instructions.

### 4.3. Epityper Mass Spectrometry

Methylation analyses were performed according to the procedure implemented by our group in a previous study [11]. MassARRAY^®^ EpiTYPER platform, with MassCleave settings, was used to determine methylation levels of *BRCA1* and *RAD51C* promoter regions. The *BRCA1* and *RAD51C* PCR primers were the same as the previous study and amplified about 400 bp of the promoters [11]. As previously described [11], two human commercial control DNA samples were used as methylated and non-methylated control samples, Human HCT116 DKO Methylated DNA and Human HCT116 DKO Non-Methylated DNA (Zymo research corporation).

### 4.4. Statistical Pipeline

Methylation data in the patients’ cohort were analyzed and classified as hypo- or hyper-methylated by comparison with methylation levels obtained in a control population, composed of 60 healthy women, as reported in Azzollini et al., 2019. Briefly, the mean methylation level of the selected CpGs within each promoter (9 CpG sites for *BRCA1* and 11 for *RAD51C*) was calculated. A thousand bootstrap samples were generated based on the distribution of the mean methylation levels in controls and the one-sided 95% percentile bootstrap confidence interval (bCI) of the controls’ mean was derived [11]. Cases were considered hypermethylated when their mean promoter methylation exceeded the one-sided 95% bCI by 0.05.

## 5. Conclusions

Considering all the above, this study confirmed that, in our cohorts, both germline and constitutional epimutations represent a very rare event in BC and OC carcinogenesis. Based on these observations, although additional data are needed to better characterize the prevalence of this phenomenon, the analysis of constitutional methylation at promoters of known cancer-predisposing genes does not seem to significantly improve the definition of cancer risk.

## Figures and Tables

**Figure 1 cancers-12-00910-f001:**
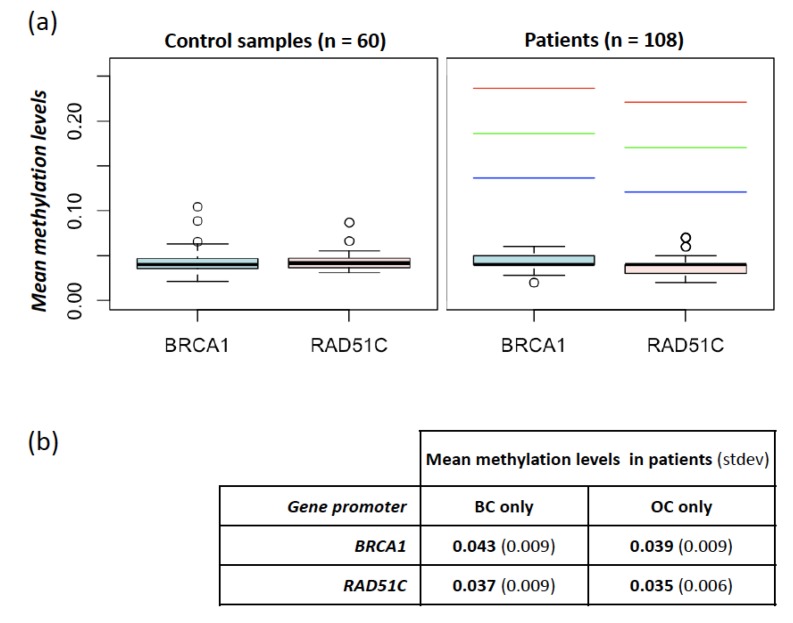
Methylation levels at *BRCA1* and *RAD51C* promoters in controls and patients. (**a**) Mean methylation levels at *BRCA1* and *RAD51C* promoters measured in 60 controls (left panel) and 108 patients (right panel). In the right panel, the red, green, and blue dashed lines indicate the thresholds used for the identification of hypermethylated cases obtained by adding the prefixed value of 0.15, 0.1, and 0.05, respectively, to the one-sided 95% bootstrap confidence interval of the controls’ mean; (**b**) mean methylation levels at *BRCA1* and *RAD51C* promoters in samples stratified according to cancer localization (breast cancer (BC) only vs. ovarian cancer (OC) only).

**Table 1 cancers-12-00910-t001:** Study cohort composition following the reported inclusion criteria. BC, breast cancer; OC, ovarian cancer; TNBC, triple negative breast cancer.

Inclusion Criteria			n. Patients
(1)	BC < 50 years	+ high mutation probability	73
(2)	OC any age	+ high mutation probability	19
(3)	TNBC < 55 years	+ high mutation probability	10
(4)	BC < 50 years/TNBC any age	+ OC any age	6

**Table 2 cancers-12-00910-t002:** Main features of breast cancer cases in the study cohort. BC, breast cancer; OC, ovarian cancer; TN, triple negative; ER, estrogen receptor, PgR, progesterone receptor; SD, standard deviation; N.A., data not available; * not in-situ lobular carcinoma.

Breast Cancer Features		BC-Only Patients	BC+OC Patients
No.		83	6
1st BC age (years)	Mean ± SD	35 ± 6	46 ± 13
	Median	33	43
	Range	24–51	36–71
1st BC Invasive	Yes	77 (93%)	5 (83%)
	No (CDIS)	2 (2%)	1 (17%)
	N.A.*	4 (5%)	0
1st BC Histotype	Ductal	66 (79%)	5 (83%)
	Lobular	4 (5%)	0
	Mixed	3 (4%)	1 (17%)
	Other	7 (8%)	0
	N.A.	3 (4%)	0
1st BC Grade	I	7 (8%)	0
	II	18 (22%)	1 (17%)
	III	30 (36%)	1 (17%)
	N.A.	28 (34%)	4 (67%)
1st BC pT	Is	1 (1%)	1 (17%)
	1	35 (42%)	1 (17%)
	2	15 (18%)	3
	3	2 (2,5%)	0
	4	2 (2,5%)	0
	N.A.	28 (34%)	1 (17%)
1st BC ER	Pos	39 (47%)	1 (17%)
	Neg	26 (31%)	1 (17%)
	N.A.	18 (22%)	4 (67%)
1st BC PgR	Pos	38 (46%)	0
	Neg	27 (32%)	2 (33%)
	N.A.	18 (22%)	4 (67%)
1st BC HER2	Pos	9 (11%)	0
	Neg	29 (35%)	2 (33%)
	N.A.	45 (54%)	4 (67%)
1st BC TN	Yes	10 (12%)	1 (17%)

**Table 3 cancers-12-00910-t003:** Main features of ovarian cancer cases in the study cohort. BC, breast cancer; OC, ovarian cancer; SD, standard deviation; N.A., data not available.

Ovarian Cancer Features		OC-Only Patients	BC + OC Patients
No.		19	6
OC Age (Years)	Mean ± SD	58 ± 12	57 ± 12
	Median	61	54.5
	Range	32–76	40–78
OC Histotype	Serous	12 (63%)	3 (50%)
	Endometrioid	3 (16%)	3 (50%)
	Undifferentiated	3 (16%)	0
	Clear cell	1 (5%)	0
OC Grade	II	5 (26%)	4 (67%)
	III	14 (74%)	2 (33%)
OC Stage	1	1 (5%)	2 (33%)
	2	1 (5%)	0
	3	5 (26%)	2 (33%)
	4	2 (11%)	0
	N.A.	10 (53%)	2 (33%)

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
