# Peer review of "Analysis of BRCA1 and RAD51C Promoter Methylation in Italian Families at High-Risk of Breast and Ovarian Cancer"

_cancers, 2020, doi:10.3390/cancers12040910_

Round 1
Reviewer 1 Report
Referee’s Comments for the Author:
The aim of this submitted manuscript was to analyze germline BRCA1 and RAD51C promoter methylation levels in BC and OC patients and compare their methylation levels with control samples using Epityper mass spectrometry. Authors declare that the analysis of constitutional methylation at promoters of these genes does not seem to substantially improve the definition of cancer risks in BC or OC patients. Although, the size of the examined cohort is a limitation of the study, this analysis is very usefull in the frame of the clinical impact of epigenetic changes on BC or OC risk. A submitted manuscript is capable of being published after a minor revision process.
Major points:
Results; Authors should include more detail analysis of BC and OC sets separately. Table or figure describing promoter methylation levels in BC and OC samples separately should be included. In addition, the next table or figure describing promoter methylation levels in different histological subtypes of BC or OC samples analyzed in this study should be included.
Materials and Methods; Authors should describe more precisely examined groups of patients. Please specify exactly, which histological subtypes and their amounts were included in the study of BC and also OC samples.
Materials and Methods; Information describing control cohort and their informed consent is missing. Please, complete this information.
Minor points:
Line 138; RAD51C instead of RAD51 gene name
Author Response
POINT TO POINT RESPONSE TO REVIEWER 1
We thank the reviewer for comments and suggestions.
As indicated by the reviewer, at Page 5, Line 137 “RAD51” was changed with “RAD51C”.
Reviewer comment: Authors should include more detail analysis of BC and OC sets separately. Table or figure describing promoter methylation levels in BC and OC samples separately should be included. In addition, the next table or figure describing promoter methylation levels in different histological subtypes of BC or OC samples analyzed in this study should be included.
Answer: Figure 1 was modified according to the reviewer’s suggestion and methylation data of BC-only cases versus OC-only cases were shown (Figure 1b). No differences were evidenced upon further stratification by histological subtypes. A new table/figure was not included since the number of cases in most subgroups was too small for adequate comparison (see the new Table 2 and Table 3). However, we specified in the text that “No differences were also noted among different histological subtypes of both BC and OC (data not shown), albeit the number of cases in each subgroup might be too small to highlight slight discrepancies.” (Page 4, Lines 117-119).
Reviewer comment: Authors should describe more precisely examined groups of patients. Please specify exactly, which histological subtypes and their amounts were included in the study of BC and also OC samples.
Answer: We added two tables (Table 2 and Table 3, Materials and methods section, Study subjects paragraph) reporting the histopathological details of BCs and OCs in the three main subgroups of patients (BC-only, OC-only and BC+OC).
Reviewer comment: Information describing control cohort and their informed consent is missing. Please, complete this information.
Answer: We included in the methods section a brief description of the control population: “The analysis was also carried out in a control population of 60 women, selected among healthy blood donors at the Immunohematology and Transfusion Medicine Service of our Institution. The age at blood withdrawal in controls ranged from 21 to 59 years (median 41) for healthy controls” (Page 8, Lines 236-238). A blank copy of the control’s informed consent is attached to this submission.
Reviewer 2 Report
This study describes assessment of BRCA1 and RAD51C promoter hypermethylation in cases of breast and ovarian cancers that do not harbor germline variants in spite of high mutation probability. The research question is worthy of investigation as loss of these repair functions as well as the epigenetic factors themselves are potential prognostic and therapeutic targets.
Suggestions:
Introduction
p.3, Line 65. This sentence is unclear. Is “emi-methylated” intended to be “hemi-methylated”?
p.3, Line 73. noticed noted
Discussion
The authors consider but mostly rule out the possibility of mosaicism, however, it might be interesting to test the BRCA1 and RAD51C promoter hypermethylation in tumor tissue from archival samples (if available).
Is there any information with regard to performance of the MassArray system compared to other methods such as microchips or methyl sequencing?
Methods
What was used for positive methylated control?
The study cohort is reasonable considering the rarity of this condition. No conclusions may be made from the subgroups, however.
Author Response
POINT TO POINT RESPONSE TO REVIEWER 2
We thank the reviewer for comments and suggestions.
In the Introduction, we made the following changes, as indicated by the reviewer:
- p.3, Line 65: “emi-methylated” was changed with “hemi-methylated”.
- p.3, Line 74: noticed was changed with “noted”
Reviewer comment: The authors consider but mostly rule out the possibility of mosaicism, however, it might be interesting to test the BRCA1 and RAD51C promoter hypermethylation in tumor tissue from archival samples (if available).
Answer: mosaicism represents an important issue since the presence of a subclone of cells with promoter hypermethylation could be masked by analyzing the methylation levels in heterogeneous samples (i.e.: blood samples). For this reason, the inclusion of the tumor tissues (FFPE specimens) represents the best approach to unmask possible candidate gene hypermethylation and demonstrate the causative effect of this epigenetic alteration.
In our previous study, carried out in a cohort of women affected with early onset breast cancer but with negative family history (Azzollini J, et al. Cancers 2019), we followed this approach after the evidence of a significant hypermethylation in the blood of a patient.
Although tumor tissues were not available for this study, our main aim was to assess the presence of germline epimutations rather than low-level constitutional or tissue-specific mosaicism. However, we agree with the reviewer that further analyses would be needed in order to explore the role of mosaic hypermethylation in patients with significant family history.
We modified the discussion section as follows: “In our analyses the lower threshold of hypermethylation was relatively high (i.e. +0.5 above the one-sided 95% bCI) and archival tumor samples were not available for this study, therefore we were able to investigate germline epimutations though we could not completely rule out the presence of mosaic epimutations.” (Page 5, Lines 171-175).
Reviewer comment:
Is there any information with regard to performance of the MassArray system compared to other methods such as microchips or methyl sequencing?
Answer: Although NGS-methyl sequencing and arrays are very robust approaches to study genome wide CpG methylation, we have used MassArray because our aim was to investigate two candidate genes, so we used a targeted approach. In addition, previous studies indicated that this platform is considered a gold standard for focused methylation assessments (Claus R, Wilop S, Hielscher T, et al. A systematic comparison of quantitative high-resolution DNA methylation analysis and methylation-specific PCR. Epigenetics. 2012;7(7):772–780).
Reviewer comment:
What was used for positive methylated control?
Answer: In the experiments, we used two human commercial control DNA, one methylated and one non-methylated (Zymo research); BRCA1 and RAD51C promoters resulted fully methylated in the methylated control and completely non-methylated in the non-methylated one, thus confirming the ability of our protocol to measure methylation changes at these regions. At Page 8, Lines 254-256, we add a sentence indicating the details of methylated and non-methyaled commercial DNA controls.